# Benzaldehyde Attenuates the Fifth Stage Larval Excretory–Secretory Product of *Angiostrongylus cantonensis*-Induced Injury in Mouse Astrocytes via Regulation of Endoplasmic Reticulum Stress and Oxidative Stress

**DOI:** 10.3390/biom12020177

**Published:** 2022-01-21

**Authors:** Kuang-Yao Chen, Yi-Ju Chen, Chien-Ju Cheng, Kai-Yuan Jhan, Lian-Chen Wang

**Affiliations:** 1Department of Parasitology, School of Medicine, China Medical University, Taichung 404, Taiwan; yiru830318@gmail.com; 2Department of Parasitology, College of Medicine, Chang Gung University, Taoyuan 333, Taiwan; d000014521@cgu.edu.tw; 3Graduate Institute of Biomedical Sciences, College of Medicine, Chang Gung University, Taoyuan 333, Taiwan; ed790101@hotmail.com; 4Molecular Infectious Disease Research Center, Chang Gung Memorial Hospital, Taoyuan 333, Taiwan

**Keywords:** excretory–secretory products, *Angiostrongylus cantonensis*, astrocytes, benzaldehydes, endoplasmic reticulum stress, oxidative stress

## Abstract

Excretory–secretory products (ESPs) are the main research targets for investigating the hosts and helminths interaction. Parasitic worms can migrate to parasitic sites and avoid the host immune response by secreting this product. *Angiostrongylus cantonensis* is an important food-borne zoonotic parasite that causes severe neuropathological damage and symptoms, including eosinophilic meningitis or meningoencephalitis in humans. Benzaldehydes are organic compounds composed of a benzene ring and formyl substituents. This compound has anti-inflammatory and antioxidation properties. Previous studies showed that 3-hydroxybenzaldehyde (3-HBA) and 4-hydroxybenzaldehyde (4-HBA) can reduce apoptosis in *A. cantonensis* ESP-treated astrocytes. These results on the protective effect underlying benzaldehyde have primarily focused on cell survival. The study was designed to investigate the molecular mechanisms of endoplasmic reticulum stress (ER stress) and oxidative stress in astrocytes in *A. cantonensis* ESP-treated astrocytes and to evaluate the therapeutic consequent of 3-HBA and 4-HBA. First, we initially established the RNA-seq dataset in each group, including normal, ESPs, ESPs + 3-HBA, and ESPs + 4-HBA. We also found that benzaldehyde (3-HBA and 4-HBA) can stimulate astrocytes to express ER stress-related molecules after ESP treatment. The level of oxidative stress could also be decreased in astrocytes by elevating antioxidant activity and reducing ROS generation. These results suggested that benzaldehyde may be a potential therapeutic compound for human angiostrongyliasis to support brain cell survival by inducing the expression levels of ER stress- and oxidative stress-related pathways.

## 1. Introduction

Angiostrongyliasis is a serious parasitic disease worldwide. It is caused by the parasitic helminth *Angiostrongylus cantonensis*. The life cycle of this nematode requires rats and mollusks [1]. Adults live in the blood vessel of rat cardiopulmonary, and then eggs become the first-stage larvae (L1). Finally, L1 are released via rat feces. After ingestion by the intermediate host, L1 can migrate to the muscle tissue and develop into infective third-stage larvae (L3). After rats eat the mollusks containing L3, the larvae can reach the host brain through blood circulation and develop into fifth-stage larvae (L5). Humans are the abnormal host for these worms through infection by ingesting infective third-stage larvae (L3) in snails, terrestrial crabs, freshwater shrimp, frogs, and fish [2,3,4,5]. This parasite can induce eosinophilic meningitis/meningoencephalitis and cause some symptoms, including fever, headache, nausea, vomiting, neck stiffness, and paraesthesia [3]. Recently, human angiostrongyliasis has been found to be an emerging disease in many areas, such as China, the USA, Europe, Australia, Jamaica, and Brazil [6,7,8,9,10,11].

The treatment of angiostrongyliasis is limited to chemical anthelmintic or immunosuppressive reagents, which are often associated with severe side effects. At present, albendazole and corticosteroids are common therapeutic drugs for eosinophilic meningitis caused by *A. cantonensis* [12,13]. Thus, the identification of more effective and safe drugs is highly necessary. *Gastrodia elata* (Tianma) is a Chinese herbal medicine that has antioxidative stress and anti-inflammation functions in humans. It has also been utilized for the treatment of headaches and dizziness [14]. 

In nature, benzaldehyde has anti-oxidation and -inflammatory functions [15,16]. 3-HBA and 4-HBA are the benzaldehydes from *G. elata.* Previous studies have demonstrated that 3-HBA has the function of anti-oxidative stress, vasculoprotection, and inflammation suppression [16,17]. On the other hand, 4-HBA is also the important bioactive constituent of *G. elata* and can promote antioxidant activation, wound repairment, and migration [18,19,20]. In protozoan parasite *Toxoplasma gondii* infection, 4-HBA inhibits *Toxoplasma gondii* growth in macrophages by inducing autophagy activation [21].

Our previous results found that the survival rate of astrocytes was increased after 3-HBA and 4-HBA treatment. Further results demonstrated that 3-HBA and 4-HBA can inhibit apoptosis-related molecule expression. Therefore, 3-HBA and 4-HBA may be the potential therapeutic molecules for human angiostrongyliasis treatment [22]. In this research, we aimed to evaluate the utility of 3-HBA and 4-HBA after ESPs treatment in astrocytes. The data demonstrated that the ESPs can cause endoplasmic reticulum stress (ER stress) and oxidative stress. Moreover, we found that 3-HBA and 4-HBA suppress ROS generation through the induction of antioxidant activation. These results suggest that benzaldehyde an effective compound for reducing oxidative stress caused by *A. cantonensis* ESPs treatment in astrocytes.

## 2. Materials and Methods

### 2.1. Ethics

The animal experiments were conducted in accordance with the standards of the Chang Gung University Institutional Animal Care and Use Committee (IACUC) in Taiwan (CGU109-198). Rats were kept in plastic cages and provided with food and water ad libitum.

### 2.2. Parasite and Experimental Infection

The lifecycle of *A. cantonensis* was maintained by *Biomphalaria glabrata* snails and Sprague–Dawley (SD) rats in a laboratory [23]. Male SD rats were purchased from the National Laboratory Animal Center, Taipei. In *A. cantonensis* infection, the L3 were collected from infected snails by digestion with 0.6% (*w/v*) pepsin-HCl (pH 2–3) for 1 h after day 21 post-infection. SD rats were inoculated with L3 by stomach intubation.

### 2.3. A. cantonensis Excretory/Secretory Product Preparation

*A. cantonensis* L5 was collected from the host brains of infected rats. Next, the worms were incubated in fetal bovine serum (FBS)-free RPMI after washing with saline, phosphate-buffered saline, ddH_2_O, and RPMI. *A. cantonensis* L5 ESPs were concentrated by centrifugal filter devices (Merck Millipore, Darmstadt, Germany). The ESP concentration was determined by a Bio-Rad Protein Assay Kit (Bio-Rad, Hercules, CA, USA), according to the instructions of the manufacturer. Finally, it was employed to treat astrocytes and then observe cell survival and gene and protein expression.

### 2.4. Astrocyte Culture

The CRL2535 (mouse brain astrocytes) was cultured at 37 °C in Dulbecco’s modified Eagle’s medium/F-12 (DMEM/F-12) (Corning, New York, NY, USA) and with 10% fetal bovine serum (FBS) (Gibco, Miami, OK, USA) in poly-L-lysine-coated culture flasks. The cells for assays were harvested from a 10 cm culture dish containing serum-free DMEM/F-12. Finally, the cells were pretreated with the drugs for 2 h and then stimulated to the ESPs of *A. cantonensis* L5.

### 2.5. Extraction of Total RNA and cDNA Synthesis

The total RNA of astrocytes cultured in medium containing ESPs and drugs was extracted as follows. TRIzol^®^ Reagent was used to isolate total RNA from different treatments. Finally, the RNA was redissolved in DEPC-treated water. Next, the iScript™ Advanced cDNA Synthesis Kit (Bio-Rad, Hercules, CA, USA) was used to generate the cDNA.

### 2.6. RNA Sequencing (RNA-Seq) Library Preparation and Sequencing

The RNA was utilized to the preparation of the sequencing library by the TruSeq Stranded mRNA Library Prep Kit (Illumina, San Diego, CA, USA) following the manufacturer’s recommendations. First, reverse transcriptase and random primers were used to generate first-strand cDNA, and the adaptors were ligated and purified with the AMPure XP system (Beckman Coulter, Beverly, MA, USA). Agilent Bioanalyzer 2100 system was employed to evaluate the library quality. Finally, the Illumina NovaSeq 6000 platform was employed to sequence the libraries (150 bp paired-end reads) (Genomics, New Taipei City, Taiwan).

### 2.7. Bioinformatics Analysis

The filtered reads were aligned and quantified by Bowtie2 (version 2.3.4.1) and RSEM (version 1.2.28). The functional analysis, including GO and KEGG, were carried out of GO terms and KEGG pathways [24,25,26,27,28,29,30].

### 2.8. Quantitative Real-Time PCR

The iQ™ SYBR^®^ Green Supermix (Bio–Rad, USA) was employed to quantitative real-time PCR on a Real-Time PCR Detection System (Bio–Rad, USA). GAPDH is an internal control.

### 2.9. Protein Extraction and Western Blotting

The cells could be disrupted in lysis buffer (RIPA buffer) containing protease inhibitors (Sigma-Aldrich, St. Louis, MO, USA). The proteins in 12% SDS-PAGE were transferred to NC membranes and incubated with antibodies against catalase (ABclonal, Woburn, MA, USA), glutathione reductase (ABclonal, Woburn, MA, USA), superoxide dismutase (ABclonal, Woburn, MA, USA), glutathione transferase kappa (ABclonal, Woburn, MA, USA), GRP78 (Proteintech, Rosemont, PA, USA), IRE1 (Signalway Antibody, Greenbelt, MD, USA), ATF6 (Proteintech, Rosemont, PA, USA), CHOP (Proteintech, Rosemont, PA, USA), PERK (Proteintech, Rosemont, PA, USA), eIF2α (Proteintech, Rosemont, PA, USA), ATF4 (Proteintech, Rosemont, PA, USA), and β-actin (Proteintech, Rosemont, PA, USA). Next, the membrane was incubated with HRP-linked secondary antibody (Sigma-Aldrich, St. Louis, MO, USA). ECL reagent (EMD Millipore, Danvers, MA, USA) and a ChemiDoc Imaging System (Bio-Rad, Hercules, CA, USA) were employed to detect and capture the immunoreactive bands. Finally, ImageJ was used to quantify the image densitometry.

### 2.10. Catalase Activity Assay

The activity of catalase in astrocytes was determined by using a catalase assay kit (Cayman, Ann Arbor, MI, USA) according to the manufacturer’s instructions. The activities were measured using a microplate reader by reading the absorbance at 540 nm.

### 2.11. Superoxide Dismutase (SOD) Activity Assay

SOD activities in astrocytes were determined by using a SOD assay kit (Cayman, USA) according to the manufacturer’s instructions. The activities were measured using a microplate reader by reading the absorbance at 440–160 nm before incubation of the plate on a shaker for 30 min.

### 2.12. Glutathione S-Transferase (GST) Activity Assay

GST activities in astrocytes were determined by using a GST assay kit (Cayman, USA) according to the manufacturer’s instructions. The activities were measured for at least 5 time points using a microplate reader by reading the absorbance at 340 nm.

### 2.13. ROS Detection

ROS generation in astrocytes was detected by using the ROS Detection Cell-Based Assay kit (Cayman, USA) according to the manufacturer’s instructions. For nuclear counterstaining, the cells were stained with DAPI. Finally, the samples were observed and photographed under a fluorescence microscope.

### 2.14. Antioxidant Capacity Assay

An antioxidant assay kit (Cayman, USA) was employed to detect antioxidant capacities in astrocytes. Controls and ESP-, 3-HBA-, and 4-HBA-treated astrocytes (1 × 10^7^ cells/mL) were homogenized or sonicated in assay buffer. Cell lysate, metmyoglobin, chromogen, and hydrogen peroxide were added and incubated in the wells on the plate. Antioxidant capacity was detected using a spectrofluorometer (Bio-Rad, USA) by reading the absorbance at 405 or 750 nm.

### 2.15. Statistical Analysis

We used Student’s *t*-test to compare the expression levels by GraphPad Prism 5 software (GraphPad, San Diego, CA, USA). Data were shown as the mean ± standard deviation. The levels of statistical significance were *p* < 0.05, *p* < 0.01, and *p* < 0.001.

## 3. Results

### 3.1. Transcriptome Profile of Mouse Astrocytes after A. cantonensis L5 ESPs or Benzaldehyde Treatment

First, we wanted to confirm the appropriate concentration of 3-HBA and 4-HBA for cell treatment in this study. Cells were pretreated with different concentrations of 3-HBA and 4-HBA (0.05, 0.1, 0.5, and 1 mM) and then treated with *A. cantonensis* L5 ESPs (250 μg/mL). The results showed that 0.1 and 0.5 mM 3-HBA and 4-HBA are the suitable concentrations for treatment by using the CCK-8 analysis (Appendix A). Next, we initially established the RNA-seq dataset in each treatment group, including normal, ESPs, ESPs + 0.1 mM 3-HBA, ESPs + 0.5 mM 3-HBA, ESPs + 0.1 mM 4-HBA, and ESPs + 0.5 mM 4-HBA. We obtained a total of 123,138,219 raw reads from polyA-tailed mRNA using the NovaSeq 6000 Sequencing System with 150 bp paired-end reads (Illumina, USA) (Table 1 and Appendix A). First, we focused on the comparative transcriptome results of normal, ESPs, ESPs + 0.5 mM 3-HBA, and ESPs + 0.5 mM 4-HBA. The transcriptome results showed that the expression of inflammation-related genes, such as serum amyloid A3, chemokines, interleukin 6, interleukin 12, and colony-stimulating factor, was obviously elevated in astrocytes after *A. cantonensis* L5 ESPs treatment. In contrast, the expression of inflammation-related genes was reduced after 3-HBA or 4-HBA treatment (Appendix A). These results suggested that benzaldehyde can reduce *A. cantonensis* L5 ESP-induced inflammation in astrocytes.

As shown in Figure 1, the differentially expressed genes in each comparison were determined by RSEM and EBSeq (PPEE <= 0.05). In total, 4753 genes were shown in the union of comparisons, including E/N, 3H/E, and 4H/E (Figure 1A–C and Appendix A). Moreover, we used a heatmap to demonstrate the expression profiles of both union and intersect sets (Figure 1D). Next, we started with functional enrichment analysis of these genes. All unigene sequences were matched against Gene Ontology (GO) terms and Kyoto Encyclopedia of Genes and Genomes (KEGG) pathways (Figure 2 and Appendix A).

### 3.2. The Expression of ER Stress-Related Molecules after 3-HBA and 4-HBA Treatment

To confirm whether 3-HBA and 4-HBA could induce ER stress-related gene expression, as shown in Figure 3, we used quantitative real-time PCR to measure the expression of genes after 3-HBA and 4-HBA treatment. The results showed that the mRNA expression levels of ER stress signaling molecules (GRP78, IRE1, ATF6, CHOP, PERK, eIF2, and ATF4) were significantly elevated after ESP treatment. Moreover, the expression level of ER stress signaling was also significantly elevated after 0.5 mM 3-HBA and 4-HBA treatment compared with ESPs alone. On the other hand, we used western blotting to detect the protein expression levels of ER stress signaling in astrocytes (Figure 4). The results showed that the protein expression levels of ER stress signaling were also significantly elevated in ESPs alone or 3-HBA and 4-HBA (0.1 and 0.5 mM) treatment. Therefore, we suggested that HBA can stimulate the activation of the ER stress signaling pathway in astrocytes after *A. cantonensis* L5 ESPs treatment.

### 3.3. The Expression of Oxidative Stress-Related Molecules after 3-HBA and 4-HBA Treatment

First, we wanted to determine whether benzaldehydes can protect astrocytes in ESPs treatment by activating antioxidants. Quantitative real-time PCR and western blotting were employed to detect the gene and protein expression of antioxidants after ESP treatment alone or 3-HBA and 4-HBA treatment (Figure 5 and Figure 6). The data showed that the expression of antioxidants (CAT, GSR, SOD, and GSTK) was significantly elevated in ESPs alone and increased in 3-HBA and 4-HBA (0.1 or 0.5 mM) treatment. The data demonstrated that HBA can induce the oxidative stress-related molecules expression after *A. cantonensis* L5 ESPs treatment.

### 3.4. The Activity of Antioxidant after 3-HBA and 4-HBA Treatment

To confirm the effect of 3-HBA and 4-HBA on the activity of antioxidants in astrocytes, we employed ELISA to examine the activity of oxidative-related enzymes, such as CAT, SOD, and GST. As shown in Figure 7, the ESPs of *A. cantonensis* L5 induced antioxidant activation in astrocytes. Moreover, we found that the activity of antioxidants was elevated after 3-HBA and 4-HBA treatment compared with ESPs alone. Therefore, we propose that HBAs reduce oxidative stress by inducing the activation of antioxidants in astrocytes.

### 3.5. 3-HBA and 4-HBA Inhibited ROS Generation after ESPs Treatment

To monitor the effect of 3-HBA and 4-HBA on ROS generation in astrocytes, we investigated whether 3-HBA and 4-HBA could reduce ESP-induced ROS generation in astrocytes (Figure 8). Cells were incubated with *A. cantonensis* L5 ESPs in the absence or presence of 3-HBA and 4-HBA, and then DHE (dihydroethidium) was added to stain ROS. Fluorescence microscopy was employed to detect ROS expression in astrocytes. The data showed that ROS expression was significantly elevated after ESPs treatment. However, 3-HBA and 4-HBA reduced ESP-induced ROS expression. The data demonstrated that HBA can inhibit oxidative stress generation through decreasing ESP-induced ROS generation.

### 3.6. 3-HBA and 4-HBA Elevate the Antioxidant Capacity after ESPs Treatment

Finally, we wanted to determine whether 3-HBA and 4-HBA could increase the antioxidant capacity in astrocytes (Figure 9). The results demonstrated that the level of antioxidant capacity was significantly higher in the cells pretreated with 3-HBA and 4-HBA than in the cells treated with ESPs alone. These data suggested that the antioxidant capacity in astrocytes can be elevated after HBA treatment.

## 4. Discussion

In the research, we determined the protective efficiency of benzaldehydes in *A. cantonensis* L5 ESPs treatment. ESPs are a mixture of helminths secreted and contain a wide range of molecules, including DNA, mRNA, microRNA, proteins, and carbohydrate [31,32]. This secretion can help parasites penetrate host barriers and reduce inflammatory responses. Moreover, ESPs also play a crucial role in reproduction and nourishment intake [33,34,35]. Because of their importance in nematode pathogenesis, ESPs are an important target in the investigation of host–parasite relationships and research in vaccine candidates and therapeutic targets. In a previous investigation, we demonstrated that *A. cantonensis* ESPs can stimulate oxidative stress, autophagy, ER stress, and cell apoptosis in astrocytes [36,37,38]. However, we used proteomic analysis to profile, identify, and characterize the proteins in the *A. cantonensis* L5 ESPs. MALDI-TOF/TOF analysis showed that a total of 51 spots were identified, such as disulfide isomerases and calreticulin. On the other hand, approximately eight immunoreactive proteins were identified, including protein disulfide-isomerase, a putative aspartic protease, and annexin [39].

In this study, we demonstrated that benzaldehydes (3-HBA and 4-HBA) can activate endoplasmic reticulum (ER) stress signaling by stimulating related molecules, including IRE1, ATF6, CHOP, PERK, eIF2, and ATF4. Moreover, our results on the expression of GRP78 revealed that benzaldehydes induce ER stress in astrocytes. The ER is the intracellular organelle for calcium (Ca^2+^) storage and has multiple functions, such as protein folding, modification, lipid biosynthesis, and membrane biogenesis [40,41]. The ER also plays a major role in the regulation of Ca^2+^ signaling, such as Ca^2+^-induced gene expression, synaptic plasticity, embryonic development, and skeletal muscle excitation–contraction coupling [42,43]. The unfolded proteins accumulation in the ER could stimulate unfolded protein response (UPR) activation, and this phenomenon is also called ER stress. Finally, this stress can induce the activation of related signaling pathways [44,45,46]. ER stress can inhibit protein synthesis and aggregation by activating chaperones, and it can also degrade abnormal proteins by the proteasome pathway. On the other hand, ER stress may increase the cell survival rate via activation of the antiapoptotic pathway [47].

ER stress activation can also lead to priming protein generation of glucose-regulated protein 78 (GRP78). GRP78 is a major marker for the evaluation of ER stress in target cells. When GRP78 is activated, GRP78 disassociates from and subsequently activates three signal transducers—IRE1, PERK and ATF6—and then induces the following pathway [48]. IRE1 can upregulate protein chaperones by activating the RNA of XBP1. PERK also activates chaperones by activating eIF2α and ATF4. Moreover, ATF6 induces cell cycle arrest by initiating gene expression programs. Finally, these mechanisms could elevate cell survival through downregulation of cell death-related molecule expression [49].

In this study, we demonstrated that oxidative stress is induced after *A. cantonensis* L5 ESP treatment in astrocytes. However, activation of antioxidants may decrease oxidative stress by inhibiting ROS generation in the treatment of 3-HBA and 4-HBA. Oxidative stress plays a significant role in the pathogenesis of parasite infection, such as *Plasmodium falciparum*, *Trypanosoma cruzi*, *Toxoplasma gondii*, *Giardia lamblia*, *Eurytrema coelomaticum*, and *Ornithodiplostomum ptychocheilus* [50,51,52,53,54,55,56,57]. After infection, oxidative stress is induced in the host and then causes cell apoptosis, inflammation, and DNA breakage. The generation of oxidative stress as a result of reactive oxygen species (ROS) (H_2_O_2_, O_2_^−^, and OH^−^) induces the impairment of antioxidant defenses [58,59]. In *A. cantonensis* infection, the expression of oxidative stress and antioxidants, including glutathione reductase (GR), glutathione peroxidase (GPx), and glutathione S-transferase (GST), was increased in the *A. cantonensis*-infected mouse central nervous system (CNS) [60,61].

In our previous study, the results showed that 3-HBA and 4-HBA can induce the protective function via regulating the generation of cell apoptosis after the treatment of ESPs [22]. At present, the therapeutic strategy for cerebral angiostrongyliasis remains controversial. Albendazole, the anthelmintics, may be effective drugs for angiostrongyliasis treatment by inhibiting nematode survival. Recent *A. cantonensis* studies have shown that the administration of anthelmintics or supportive therapy with corticosteroids is also in dispute. In our previous study, we found that administration of albendazole and dexamethasone has effective utility for cerebral angiostrongyliasis by reducing clinical symptoms and pathological changes, including meningitis, encephalitis, hemorrhage, and worm survival [13]. The results showed that the combination of albendazole with dexamethasone is the most effective therapy for cerebral angiostrongyliasis. Moreover, this therapeutic strategy could reduce inflammation in the CNS. For example, the cytokine expression, such as of IL-1β, IL-4, and IL-5, was decreased in the CNS after drug treatment [62].

## 5. Conclusions

In conclusion, we demonstrated that *A. cantonensis* L5 ESPs induce ER stress and oxidative stress in mouse astrocytes. However, we found that 3-HBA and 4-HBA can reduce ROS generation by stimulating antioxidant activation, which results in decreased oxidative stress. According to our findings, benzaldehydes may be potential molecules for the therapy of angiostrongyliasis. 

## Figures and Tables

**Figure 1 biomolecules-12-00177-f001:**
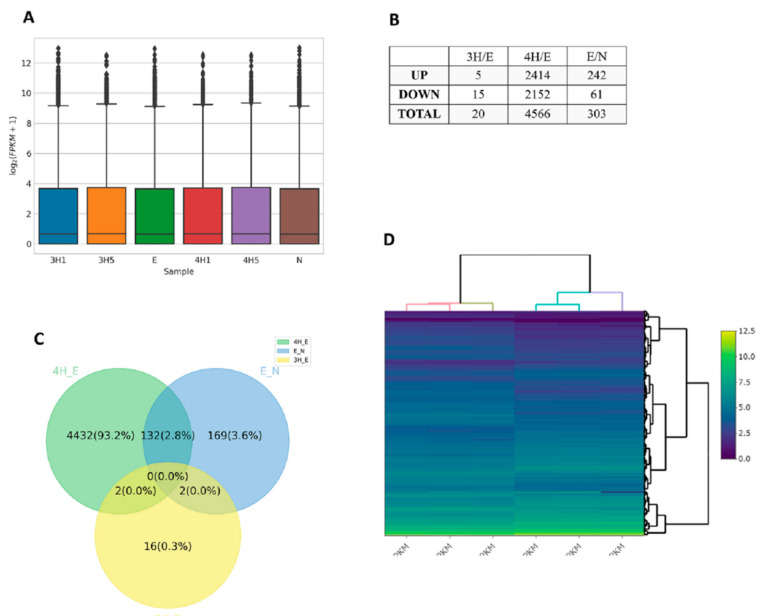
Summary of the transcriptome profile of mouse brain astrocytes after excretory/secretory product or benzaldehyde treatment. (**A**) Boxplot of FPKM distribution among samples. N: normal, E: 250 μg/mL ESPs treatment, 3H1: ESPs + 0.1 mM 3-HBA, 3H5: ESPs + 0.5 mM 3-HBA, 4H1: ESPs + 0.1 mM 4-HBA, 4H5: ESPs + 0.5 mM 4-HBA. (**B**) Number of DEGs in comparison sets E/N, 3H/E, and 4H/E. (**C**) Venn diagram was used to show the number of differentially expressed genes in each comparison determined by RSEM and EBSeq (PPEE <= 0.05). A total of 4753 genes were shown in the union of comparisons. (**D**) A heatmap was employed to demonstrate the expression profiles of both union and intersect sets.

**Figure 2 biomolecules-12-00177-f002:**
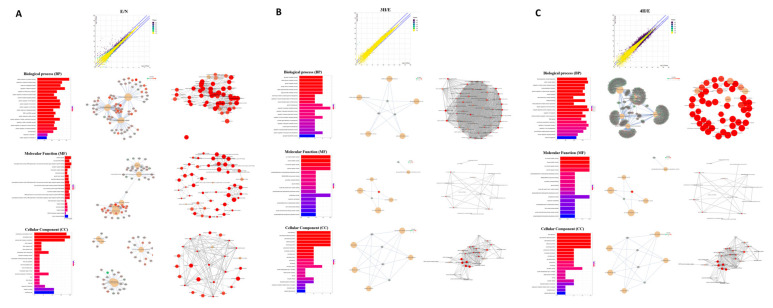
GO annotations of unigenes in transcriptome profile. (**A**) E compared to N. (**B**) 3H compared to E. (**C**) 4H compared to E.

**Figure 3 biomolecules-12-00177-f003:**
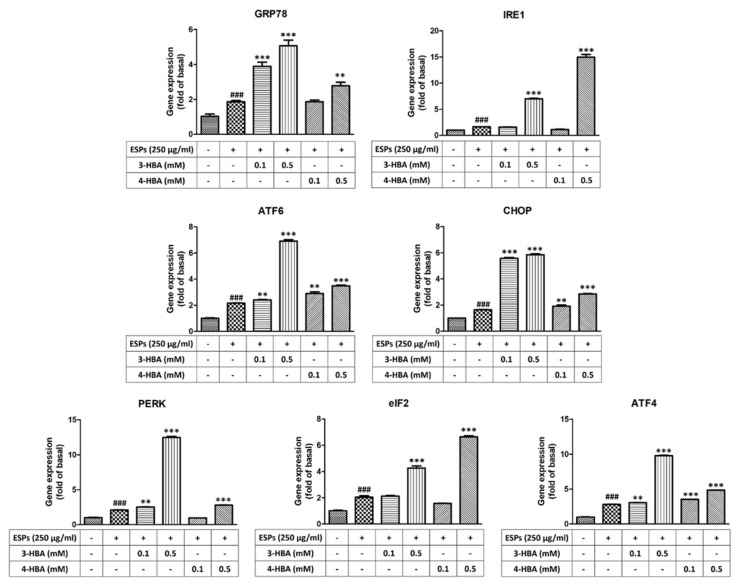
3-HBA and 4-HBA induce the expression of ER stress-related genes in ESP-treated astrocytes. Cells were pretreated with 3-HBA and 4-HBA (0.1 and 0.5 mM) for 4 h and then incubated with 250 μg/mL ESPs for 12 h. mRNA expression was determined by real-time qPCR (*n* = 3). + and - represent cells were pretreated or not with ESPs, 3-HBA or 4-HBA. *^###^ p* < 0.001, compared to control. *** p* < 0.01, **** p* < 0.001, compared to cells exposed to ESPs.

**Figure 4 biomolecules-12-00177-f004:**
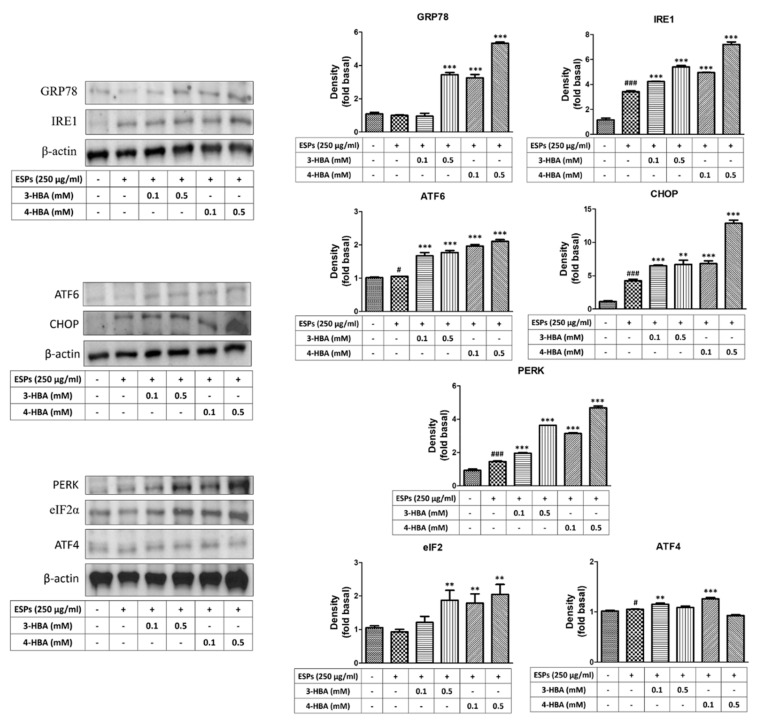
3-HBA and 4-HBA induced the expression of ER stress-related proteins in ESP-treated astrocytes. Cells were pretreated with 3-HBA and 4-HBA (0.1 and 0.5 mM) for 4 h and then incubated with 250 μg/mL ESPs for 12 h. Protein expression levels were determined by western blotting (*n* = 3). + and - represent cells were pretreated or not with ESPs, 3-HBA or 4-HBA. *^#^ p* < 0.05, *^###^ p* < 0.001, compared to control. *** p* < 0.01, **** p* < 0.001, compared to cells exposed to ESPs.

**Figure 5 biomolecules-12-00177-f005:**
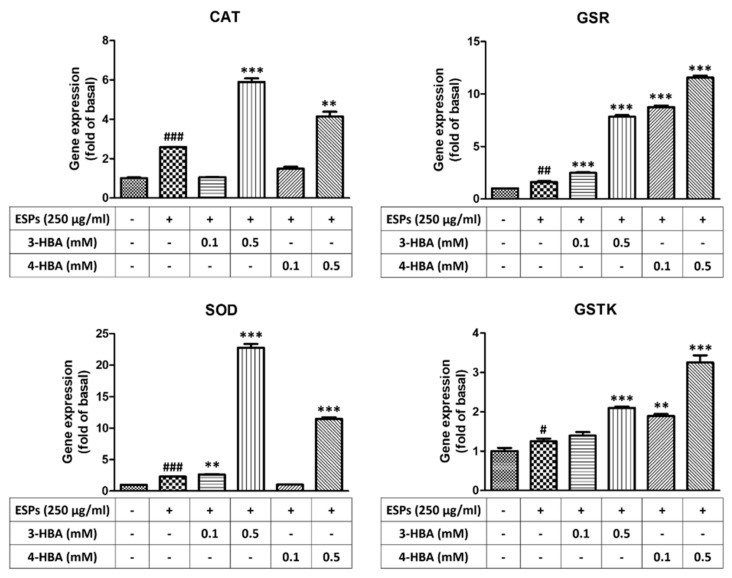
3-HBA and 4-HBA induced the gene expression of antioxidants in ESP-treated astrocytes. Cells were pretreated with 3-HBA and 4-HBA (0.1 and 0.5 mM) for 4 h and then incubated with 250 μg/mL ESPs for 12 h. mRNA expression was determined by real-time qPCR (*n* = 3). + and - represent cells were pretreated or not with ESPs, 3-HBA or 4-HBA. *^#^ p* < 0.05, *^##^ p* < 0.01, *^###^ p* < 0.001, compared to the control. *** p* < 0.01, **** p* < 0.001, compared to cells exposed to ESPs.

**Figure 6 biomolecules-12-00177-f006:**
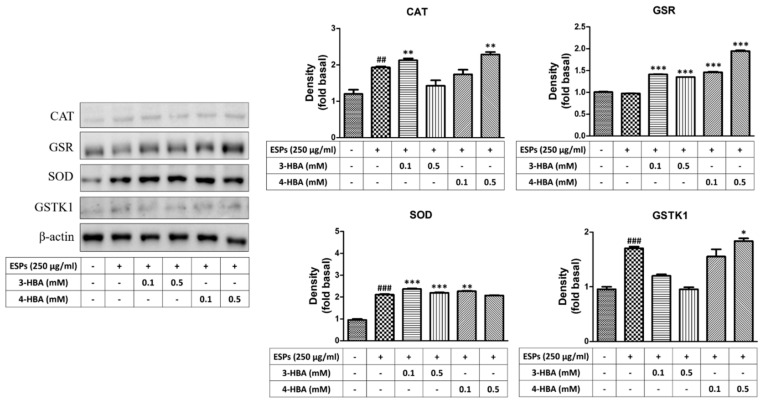
3-HBA and 4-HBA induced the protein expression of antioxidants in ESP-treated astrocytes. Cells were pretreated with 3-HBA and 4-HBA (0.1 and 0.5 mM) for 4 h and then incubated with 250 μg/mL ESPs for 12 h. Protein expression levels of antioxidants were detected by western blotting (*n* = 3). + and - represent cells were pretreated or not with ESPs, 3-HBA or 4-HBA. *^##^ p* < 0.01, *^###^ p* < 0.001, compared to control. ** p* < 0.05, *** p* < 0.01, **** p* < 0.001, compared to cells exposed to ESPs.

**Figure 7 biomolecules-12-00177-f007:**
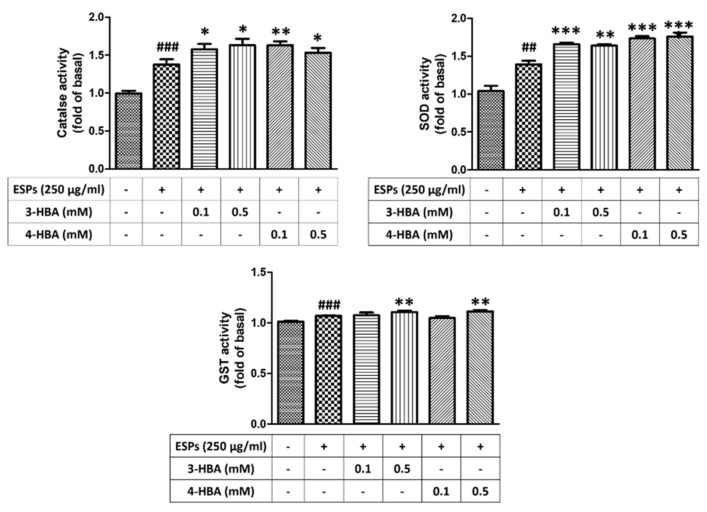
3-HBA and 4-HBA induced antioxidant activation in ESP-treated astrocytes. Cells were pretreated with 3-HBA and 4-HBA (0.1 and 0.5 mM) for 4 h and then incubated with 250 μg/mL ESPs for 12 h. Activities of catalase, SOD, and GST were determined by activity assay (*n* = 3). *^##^ p* < 0.01, *^###^ p* < 0.001, compared to control. + and - represent cells were pretreated or not with ESPs, 3-HBA or 4-HBA. ** p* < 0.05, *** p* < 0.01, **** p* < 0.01, compared to cells exposed to ESPs.

**Figure 8 biomolecules-12-00177-f008:**
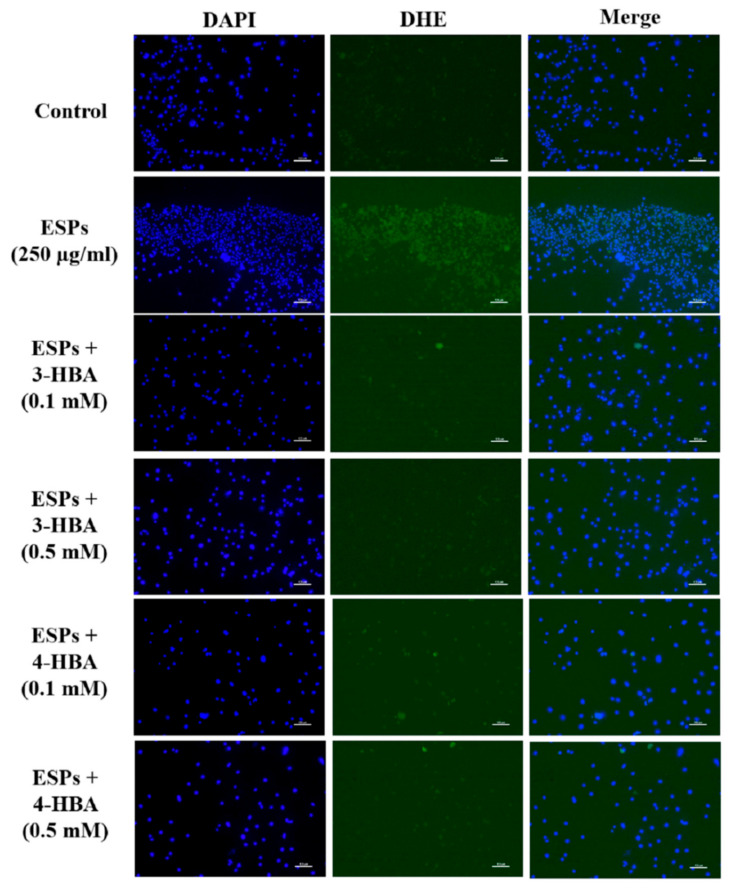
3-HBA and 4-HBA reduced ROS generation in ESP-treated astrocytes. Cells were pretreated with 3-HBA and 4-HBA (0.1 and 0.5 mM) for 4 h and then incubated with 250 μg/mL ESPs for 12 h. An ROS detection kit was used to detect ROS generation by fluorescence microscopy (*n* = 3). Scale bar = 100 µm.

**Figure 9 biomolecules-12-00177-f009:**
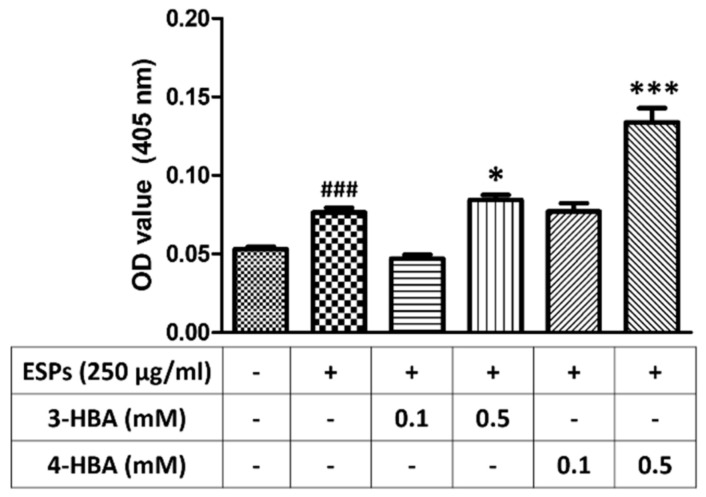
Elevation of antioxidant capacity in astrocytes treated with 3-HBA and 4-HBA. Cells were pretreated with 3-HBA and 4-HBA (0.1 and 0.5 mM) for 4 h and then incubated with 250 μg/mL ESPs for 12 h. An antioxidant assay kit was employed to detect antioxidant capacities in astrocytes (*n* = 3). + and - represent cells were pretreated or not with ESPs, 3-HBA or 4-HBA. *^###^ p* < 0.001, compared to control. ** p* < 0.05, **** p* < 0.001, compared to cells exposed to ESPs.

**Table 1 biomolecules-12-00177-t001:** Characteristics of the transcriptome of mouse brain astrocytes after excretory/secretory product or benzaldehyde treatment.

Sample_Name	GC.	Length	Length_Mean	Phred	Q20_Ratio	Q30_Ratio	Qual_Mean	Read_Counts	Total_Bases
N_R1	48.65%	20–151	144.32	33	99.04%	96.29%	36.42	25,920,999	3,740,844,537
N_R2	48.78%	20–151	143.69	33	98.43%	94.35%	36.10	25,920,999	3,724,577,535
E_R1	48.00%	20–151	144.66	33	98.97%	96.05%	36.38	23,617,944	3,416,535,104
E_R2	48.22%	20–151	144.22	33	98.61%	94.87%	36.19	23,617,944	3,406,183,730
3H1_R1	48.48%	20–151	144.94	33	99.00%	96.18%	36.4	23,470,406	3,401,781,446
3H1_R2	48.68%	20–151	143.85	33	98.28%	94.04%	36.04	23,470,406	3,376,330,842
3H5_R1	49.22%	20–151	145.23	33	99.03%	96.28%	36.42	23,669,860	3,437,686,443
3H5_R2	49.38%	20–151	144.55	33	98.42%	94.38%	36.1	23,669,860	3,421,432,553
4H1_R1	48.82%	20–151	144.87	33	98.99%	96.18%	36.4	27,305,448	3,955,655,684
4H1_R2	48.97%	20–151	143.86	33	98.31%	94.10%	36.06	27,305,448	3,928,240,745
4H5_R1	49.20%	20–151	145.02	33	99.02%	96.25%	36.41	25,074,561	3,636,192,527
4H5_R2	49.37%	20–151	144.24	33	98.32%	94.09%	36.06	25,074,561	3,616,740,196

## Data Availability

All data generated or analyzed during this study are available from the corresponding author upon reasonable request.

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
