# Peer review of "Benzaldehyde Attenuates the Fifth Stage Larval Excretory–Secretory Product of *Angiostrongylus cantonensis*-Induced Injury in Mouse Astrocytes via Regulation of Endoplasmic Reticulum Stress and Oxidative Stress"

_biomolecules, 2022, doi:10.3390/biom12020177_

Round 1

Reviewer 1 Report

In the present manuscript, Wang et al. investigate the mechanisms underlying their previous observation regarding the therapeutic potential of Benzaldehydes (3-HBA and 4-HBA) in case of Angiostrongyliasis caused by Angiostrongylus cantonensis. The authors argue that anti-inflammatory and anti-antioxidation properties of 3-HBA and 4-HBA underlie the protective mechanism. Using RNA-seq analyses, the authors propose that in response to HBA treatment, astrocytes initiate signaling related to ER and oxidative stress, and that these responses are needed for the protective mechanisms.

On the whole, the authors provide solid observation, and the manuscript is well-written. However, there are concerns that would need to be addressed for the manuscript to be ready for publication.

  • Authors should mention the rationale of choosing the different concentration of 3-HBA and 4-HBA.

  • It would be worthwhile to assess the beneficial effects of combined 3-HBA and 4-HBA treatment.

  • In the legend for Fig 8, the authors indicate incubation with 250ug/ml of ESPs, but figure shows addition of 500ug/ml. The authors should fix this error.

DHE staining images with 0.1 mM 3-HBA and 4-HBA are also missing in the figure, despite being mentioned in the legend.

The data shown in Fig. 9 should also be quantified.

  • It is recommended that the authors provide full Western blot images for the data shown in Figs. 4 and 6.

  • The authors should provide a rationale for why the cells are pre-treated with the HBA compounds? What would be the effects of adding 3-HBA or 4-HBA acutely after 12 h ESP incubation?

  • Have the authors performed a dose-response curve for the drugs? This should either be mentioned or referenced.

  • Gene expression and protein abundance in Figs. 5 and 6 do not fully match the results of the activity assays shown in Fig. 7. What could be reason for these data?

  • Font size of the text in Fig 2 (GO analyses) is too small to be able to read the various labels.

Author Response

Thanks to the comment of reviewer. We response your question and suggestion in the word file (response for Reviewer 1).

Reviewer 2 Report

In this manuscript Chen et al., examine the effects of 3-HBA and 4-HBA on A. cantonensis

ESPs treatment in astrocytes. Authors propose that 3-HBA and 4-HBA exacerbate ER stress and alleviate ROS generation in response to L5 ESP in astrocytes. In this respect, they used RNA-seq approach in addition they showed both 3-HBA and 4-HBA induce expression and enhance activity of antioxidant such as CAT, SOD, and GST. The rationale and several parts of the paper, however, need clarification and additional experiments are critically required to support the claims made by the authors. My main concerns and comments follow:

Specific concerns:

  • The authors argue that 3-HBA and 4-HBA treatment reduces ESP induced ROS generation by enhances expression and activity of antioxidant genes like CAT, SOD and GST. In contrast, it seems from results that ESP itself also significantly enhances expression and activity of antioxidant genes like CAT, SOD and GST. How do authors explain this?
  • In Figure 1B and C, the up and down regulated genes in 3H/E, and 4H/E, please mention in legends the concentration of 3-HBA and 4-HBA that was used to get this analysis. Does it includes both the concentration combined 3H1 and 3H5 as shown in figure 1A?
  • In line 226-228, the authors state that “The data showed that the expression of antioxidants (CAT, GSR, SOD, and GSTK) was significantly elevated in ESPs alone and increased in 3-HBA and 4-HBA (0.1 mM) treatment “ but the in results it seems that 0.1mM was not sufficient to induce antioxidant expression (figure 5 and figure 6 ).
  • To support their claim “that HBA can inhibit oxidative stress generation through decreasing ESPs-induced ROS generation” authors should utilize additional quantitative and more robust approaches to measure superoxide generation and ROS generation. Authors should also measure H2O2 generation by Amplex red assay. Also, use of a positive control is strongly recommended.
  • Does ESP also induce expression/activation of NOX which causes ROS generation or by other ROS sources mitochondrial which are blocked by 3-HBA and 4-HBA?
  • It is still unclear and difficult to conclude from the experiments performed that how 3 -HBA and 4-HBA inhibits ESP induced ROS generation.

Additional concerns:

  • The labelling in Figure 1D seems missing.
  • Figure 2, the entire quality of figures is very poor and blurry making it difficult to understand. Please modify/edit the quality of figures.
  • Authors should also confirm by qPCR inflammation-related genes such as serum amyloid A3, chemokines, interleukin 6, interleukin 12, and colony stimulating factor (as mentioned in line 171-175) in ESP and 3-HBA and 4-HBA treated samples.
  • It would be nice to provide the list of sequence of primers used in the study.

Author Response

Thanks to the comment of reviewer. We response your question and suggestion in the word file (response for Reviewer 2).

Round 2

Reviewer 1 Report

The authors have addressed all my concerns.

Author Response

Thanks to the comment of reviewer.

Reviewer 2 Report

In the manuscript Chen et al., claim that Benzaldehydes 3-HBA and 4-HBA are effective compounds that may have therapeutic potential to support brain cell survival by inducing the expression of ER stress and suppressing ESP induced ROS generation. However, the present study lacks several crucial experiments to strongly support their claim.

  • One of the main findings that 3-HBA and 4-HBA attenuates ESP induced ROS generation is merely supported non-quantitative microscopy images. To strengthen their claim authors should employ more quantitative approaches.
  • Additionally, in page 10, there is an error in microscopic image showing ESP+4HBA (0.1 mM) and ESP+4HBA (0.5 mM) since they look very similar.
  • Authors show that ESP induces ER stress which is enhanced by HBA treatment. In figure 4, the western blot shows interesting increase in IRE1 expression. Authors should also analyze the activation of the various branches of UPR by checking phosphorylation of IRE1 and PERK.
  • Previous reports (Martinon et al., Nat Immunol. 2010 May; 11(5): 411–418.) indicates that IRE1-XBP1 signaling also regulated inflammatory cytokines like IL6, IL1b And since authors find changes in inflammatory cytokine production (Page 4 line 178-179, “The transcriptome results showed that the expression of inflammation-related genes, such as serum amyloid A3, chemokines, interleukin 6..”), it would be relevant to check IRE1a phosphorylation and XBP1s expression in response to 3HBA and 4-HBA in ESP treated cells.
  • In Figure 5, 6 and 7, authors show increase in catalase activity. Aothurs should also analyze H2O2 levels in their study.

Author Response

Thanks to the comment of reviewer.

Round 3

Reviewer 2 Report

The present study by Chen et al., warrants additional suggested experiments to be performed that are crucial to strongly supporting their claim before considering the manuscript for publication. The suggested experiment are relevant for the study and would considerably quality of manuscript.